# DISCRETE FLOW POSTERIORS FOR VARIATIONAL INFERENCE IN DISCRETE DYNAMICAL SYSTEMS

## ABSTRACT

Each training step for a variational autoencoder (VAE) requires us to sample from the approximate posterior, so we usually choose simple (e.g. factorised) approximate posteriors in which sampling is an efficient computation that fully exploits GPU parallelism. However, such simple approximate posteriors are often insufficient, as they eliminate statistical dependencies in the posterior. While it is possible to use normalizing flow approximate posteriors for continuous latents, there is nothing analogous for discrete latents. The most natural approach to model discrete dependencies is an autoregressive distribution, but sampling from such distributions is inherently sequential and thus slow. We develop a fast, parallel sampling procedure for autoregressive distributions based on fixed-point iterations which enables efficient and accurate variational inference in discrete state-space models. To optimize the variational bound, we considered two ways to evaluate probabilities: inserting the relaxed samples directly into the pmf for the discrete distribution, or converting to continuous logistic latent variables and interpreting the K-step fixed-point iterations as a normalizing flow. We found that converting to continuous latent variables gave considerable additional scope for mismatch between the true and approximate posteriors, which resulted in biased inferences, we thus used the former approach. We tested our approach on the neuroscience problem of inferring discrete spiking activity from noisy calcium-imaging data, and found that it gave accurate connectivity estimates in an order of magnitude less time.

The development of variational auto-encoders (VAE) (Kingma & Welling, 2014; Rezende et al., 2014) has enabled Bayesian methods to be applied to very large-scale data, by leveraging neural networks, trained by stochastic gradient ascent, to approximate the posterior. Importantly, to perform each stochastic gradient ascent step, we need to sample from the current approximate posterior, restricting us to approximate posteriors in which sampling can be performed rapidly by leveraging parallel GPU computations. As such, most work on VAE's has used factorised approximate posteriors (e.g. Kingma & Welling, 2014; Rezende et al., 2014; Blundell et al., 2015; Higgins et al., 2017; Aitchison et al., 2017), but in many domains of interest we expect the posterior over latents to be highly correlated, not only because the posterior inherits correlations from the prior (e.g. as we might find in a dynamical system), but also because the likelihood itself can induce correlations due to effects such as explaining away (Pearl, 1988a;b). One approach to introducing correlations into the approximate posterior is normalizing flows (Rezende & Mohamed, 2015; Kingma et al., 2016) which transforms variables generated from a simple, often factorised distribution into a complex correlated distribution, in such a way that the determinant of the Jacobian (and thus the probability of the transformed variables) can easily be computed.

However, the normalizing flow approach can only be applied to continuous latents, and there are important problems which require discrete latent variables and correlated posteriors, making efficient and accurate stochastic variational inference challenging. In particular, we consider the neuroscience problem of inferring the correlated spiking activity of neural populations recorded by calcium imaging. Due to the indirect nature of calcium imaging, spike inference algorithms must be used to infer the underlying neural spiking activity (Friedrich et al., 2017; Speiser et al., 2017). Not only does the data naturally give strong explaining away induced anticorrelations (as a spike in nearby timebins produces very similar data), but there are prior correlations induced by synaptic connectivity which induces similar correlations in the approximate posterior.

To address these challenging tasks with discrete latent variables, and correlated approximate posteriors, we considered two approaches. First, we considered applying normalizing flows by transforming our discrete latents into continuous latents, which are thresholded to recover the original discrete variables (Maddison et al., 2016). However, we found that working with continuous variables gave rise to far more scope for mismatch between the approximate and true posteriors than working with discrete variables, and that this mismatch resulted in biased inferences. Instead, we developed a fast-sampling procedure for discrete autoregressive posteriors. In particular, we considered an autoregressive approximate posterior and circumvent the requirement for slow sequential sampling by developing flow-like fixed-point iterations that are guaranteed to sample the true posterior after $T$ iterations, but in practice converge much more rapidly (in our simulations, $\sim 5$ iterations), and efficiently exploit GPU parallelism.

Applying the flow-like fixed point iterations to simulated neuroscience problems, we were able to sample from autoregressive approximate posteriors in almost the same time required for factorised posteriors, and at least an order of magnitude faster than sequentially sampling from the underlying autoregressive process, allowing us to realize the benefits of correlated posteriors in large-scale settings.

# 1 BACKGROUND

The evidence lower bound objective (ELBO), takes the form of an expectation computed over the approximate posterior, $Q_\phi(z)$,

$$\mathcal{L} = E_{Q_\phi(z)} \left[ \log P_\theta(x|z) + \log P_\theta(z) - \log Q_\phi(z) \right] \tag{1}$$

Optimizing this objective with respect to the generative model parameters, $\theta$, is straightforward, as we can push the derivative inside the expectation, and perform stochastic gradient ascent. However, we cannot do the same with the recognition parameters, $\phi$, as they control the distribution over which the expectation is taken. To solve this issue, the usual approach is the reparameterisation trick (Kingma & Welling, 2014; Rezende et al., 2014) which performs the expectation over IID random noise, and transforms this noise into samples from the approximate posterior,

$$\mathcal{L} = E_\epsilon \left[ \log P_\theta(x, z(\epsilon; \phi)) - \log Q_\phi(z(\epsilon; \phi)) \right]. \tag{2}$$

As the recognition parameters $\phi$ no longer appear in the distribution over which the expectation is taken, we can again optimize this expression using stochastic gradient descent.

While the reparameterisation trick is extremely effective for continuous latent variables, it cannot be used for discrete latents, as we cannot back-propagate gradients through discrete $z(\epsilon; \phi)$. To rectify this issue, one approach is to relax the discrete variables, $\bar{z}$, to form an approximately equivalent model with continuous random variables, $\hat{z}$, through which gradients can be propagated (Maddison et al., 2016; Jang et al., 2016). To consider the simplest possible case, we take a single binary variable, $\bar{z}$, drawn from a Bernoulli distribution with log-odds ratio $u$, and probability $p = \sigma(u) = 1/(1 + e^{-u})$,

$$\bar{P}(\bar{z}) = \text{Bernoulli}(\sigma(u)). \tag{3}$$

Instead of sampling $\bar{z}$ directly from a Bernoulli, we can obtain samples of $\bar{z}$ from the same distribution by first sampling from a Logistic, and thresholding that sample using,

$$l = \text{Logistic}(u, 1), \qquad \bar{z} = \Theta(l) = \lim_{\beta \to \infty} \sigma(\beta l), \qquad \hat{z} = \sigma(\beta l), \tag{4}$$

where $\Theta(l)$ is the Heaviside step function, which is 0 for negative inputs, and 1 for positive inputs, and $\hat{z}$ is a relaxed version of the original discrete latent variable that lies between 0 and 1 and becomes equal to $\bar{z}$ in the limit that the inverse-temperature, $\beta$, goes to infinity.

We now have two options for how we compute probaiblities in the VAE setting for relaxed discrete latent variables.

## 1.1 EVALUATING PROBABILITIES UNDER THE DISCRETE MODEL

The most straightforward approach is to simply insert the relaxed variables, $\hat{z}$ into the original probability mass function for the discrete model, $\bar{P}(\hat{z})$, and $\bar{Q}(\hat{z})$ (Jang et al., 2016). Taking a univariate

example, this gives,

$$\log \bar{\mathrm{P}}\left(\hat{z}\right) = \hat{z} \log p + (1 - \hat{z}) \log(1 - p). \tag{5}$$

However, this immediately highlights a key issue: to obtain a valid variational bound, we need to evaluate the probability density of samples from the approximate posterior. In our case, samples from the approximate posterior are the relaxed variables, $\hat{z}$, so we need to evaluate $\hat{\mathrm{Q}}\left(\hat{z}\right)$. However, we are actually using a different expression, $\bar{\mathrm{Q}}\left(\hat{z}\right)$, and while this may in practice be an effective approximation, it cannot give us a valid variational bound. To obtain a valid variational bound, we must use $\bar{\mathrm{Q}}\left(\bar{z}\right)$, which we can compute but not differentiate.

## 1.2 Evaluating probabilities under the continuous model

To obtain a valid variational bound, we need to evaluate the actual probability of the relaxed variable, i.e. we need to compute $\hat{\mathrm{Q}}\left(\hat{z}\right)$, and $\hat{\mathrm{P}}\left(\hat{z}\right)$. While we could work with the $\hat{z}$ directly, this is known to be numerically unstable so instead we work in terms of the logistic variables, $l$ (Maddison et al., 2016). These two approaches are equivalent, in the sense that the ratio of prior and approximate posterior probabilities is the same, because the gradient terms introduced by the change of variables cancel. The generative model is described above (Eq. 4), but we have not yet specified the approximate posterior, $\hat{\mathrm{Q}}\left(l\right)$. Following usual practice Maddison et al. (2016), we use a Logistic for the approximate posterior, with learned mean and scale parameter 1 (so as to avoid a problem described below in Fig. 1E). Further, note that we have not specified whether these distributions correspond to the discrete or relaxed model, as we can obtain either $\hat{z}$ or $\bar{z}$ from exactly the same $l$, by using different inverse-temperatures for the transformation (Eq. 4).

## 2 Results

We are interested in discrete dynamical systems with autoregressive generative models and approximate posteriors given by,

$$\bar{\mathrm{P}}\left(\bar{\mathbf{z}}_t | \bar{\mathbf{z}}_{1:t-1}\right) = \mathrm{Bernoulli}\left(\boldsymbol{\sigma}\left(\mathbf{u}\left(\bar{\mathbf{z}}_{1:t-1}\right)\right)\right), \tag{6a}$$

$$\bar{\mathrm{Q}}\left(\bar{\mathbf{z}}_t | \mathbf{x}, \bar{\mathbf{z}}_{1:t-1}\right) = \mathrm{Bernoulli}\left(\boldsymbol{\sigma}\left(\mathbf{v}\left(\mathbf{x}, \bar{\mathbf{z}}_{1:t-1}\right)\right)\right). \tag{6b}$$

These equations can be rewritten using continuous Logistic variables,

$$\hat{\mathrm{P}}\left(\boldsymbol{l}_t | \boldsymbol{l}_{1:t-1}\right) = \mathrm{Logistic}\left(\mathbf{u}\left(\boldsymbol{\sigma}\left(\beta \boldsymbol{l}_{1:t-1}\right)\right), 1\right) \tag{7a}$$

$$\hat{\mathrm{Q}}\left(\boldsymbol{l}_t | \boldsymbol{l}_{1:t-1}\right) = \mathrm{Logistic}\left(\mathbf{v}\left(\mathbf{x}, \boldsymbol{\sigma}\left(\beta \boldsymbol{l}_{1:t-1}\right)\right), 1\right) \tag{7b}$$

where $\mathbf{u}$ and $\mathbf{v}$ are map from past activations to the logits value for the next time-step. Note that it is not possible to use classical message passing techniques to perform inference in this model due to the relatively long temporal dependencies.

## 2.1 A flow-like sampling procedure for discrete autoregressive distributions

One approach to sampling the sequential autoregressive posterior in Eq.(7b) is to use,

$$\boldsymbol{l}_t = \boldsymbol{\eta}_t + \mathbf{v}\left(\mathbf{x}, \boldsymbol{\sigma}\left(\beta \boldsymbol{l}_{1:t-1}\right)\right) = \mathbf{f}_t(\boldsymbol{l}_{1:t-1}), \tag{8}$$

where $\boldsymbol{\eta}_t$ is $\mathrm{Logistic}\left(0, 1\right)$ noise, $\boldsymbol{l}_t$ and $\boldsymbol{\eta}_t$ are length $N$ vectors, and we define $\mathbf{f}_t$ so as to highlight the temporal dependencies (cf. Eq.10). However, in practice these iterations can be extremely slow as the sequential structure (Fig. 1A top) fails to fully exploit the parallelism available on today's GPU hardware. As such, we considered fixed-point iterations that do fully utilize GPU parallelism. To construct these iterations, we simply apply the autoregressive updates at all time-steps in parallel, (rather than sequentially, see Fig. 1A),

$$\boldsymbol{l}^{k+1}(\boldsymbol{\eta}, \boldsymbol{l}^k) = \boldsymbol{\eta} + \mathbf{v}\left(\mathbf{x}, \boldsymbol{\sigma}\left(\beta \boldsymbol{l}^k\right)\right). \tag{9}$$

where all the variables are $N \times T$ matrices, and where $\mathbf{v}\left(\mathbf{x}, \boldsymbol{\sigma}\left(\beta \boldsymbol{l}^k\right)\right)$ is simply Eq. (8) computed in parallel across time for an externally specified input. Making the time dependencies explicit,

$$\boldsymbol{l}_t^{k+1} = \boldsymbol{\eta}_t + \mathbf{v}_t\left(\mathbf{x}, \boldsymbol{\sigma}\left(\beta \boldsymbol{l}_{1:t-1}^k\right)\right) = \mathbf{f}_t(\boldsymbol{l}_{1:t-1}^k). \tag{10}$$

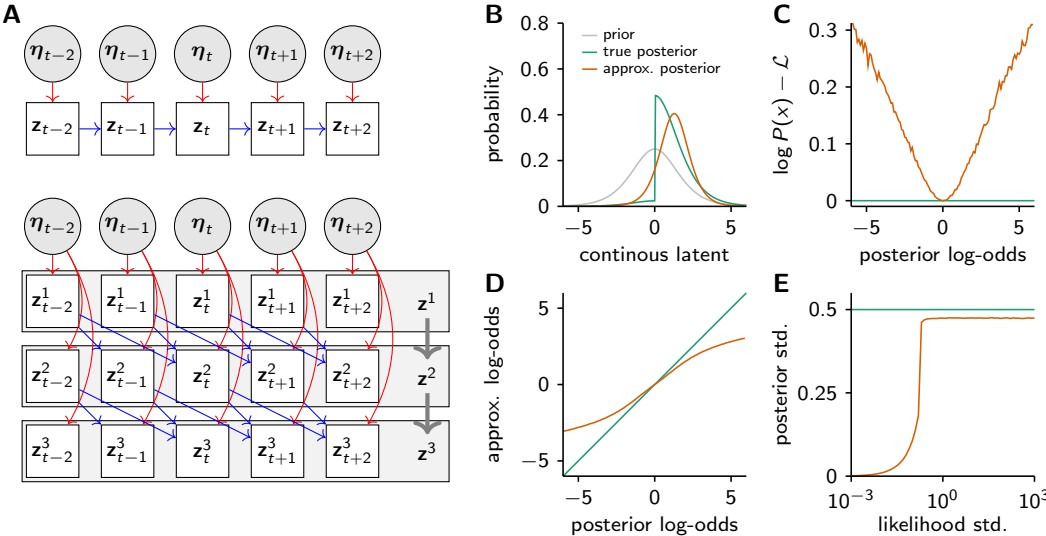

Figure 1: Differences between autoregressive and flow posteriors, based on discrete or continuous latents. **A** The sequential autoregressive (top), and parallel flow-like sampling procedure (bottom). **B** The true and approximate posteriors for continuous Logistic latent variables. **C** The difference between the model evidence and the ELBO induced by the posterior mismatch in **B**. **D** Biased inferences induced by the mismatch between the prior and approximate posterior. **E** The standard deviation of $\hat{z}$ as we modulate the standard deviation of a Gaussian likelihood encouraging $\hat{z} \approx 0.5$.

To show that this will become equivalent to the autoregressive model after $K = T$ iterations, note that after the first fixed-point iteration, $l_1^1$ is exactly equal to the sequential autoregressive result, $l_1$ (Eq. 8), as $l_1^1 = \mathbf{f}_1(\emptyset) = l_1$, because there is no history to take into account (which we represent, in a slight abuse of notation, by the empty set). After the second fixed-point iteration, $l_1^2$ and $l_2^2$ are equal to the corresponding sequential autoregressive values, as $l_1^2$ is still equal to the correct, autoregressive value, (i.e. $l_1^2 = \mathbf{f}_1(\emptyset) = l_1$), and $l_2^2 = f_2(l_1^1) = f_2(l_1) = l_2$. By induction, after $t$ iterations, the first $t$ time-points $l_{1:t}^t$ are now equal to their sequential autoregressive values, $l_{1:t}^t = l_{1:t}$ until after $T$ parallel iterations we have equality for the whole time-series.

Thus, the worst-case is extremely slow, converting a $\mathcal{O}(T)$ computation to an order $\mathcal{O}(T^2)$ computation. However, in practice the iterations reach steady-state rapidly so we are able to use $K \ll T$, giving an order of magnitude improvement in efficiency by making improved use of GPU parallelism. Finally, note that this procedure generalizes straightforwardly to the case of Categorical variables: we convert the sigmoid to a softmax, and convert the Logistic to a Gumbel distribution.

While this section gives a fast procedure for sampling from the approximate posterior, it is still unclear how we should evaluate probabilities. We have two options, which mirror the two options introduced above (Sec. 1.1 and Sec. 1.2 respectively).

## 2.2 EVALUATING PROBABILITIES UNDER THE DISCRETE MODEL

First, we can compute relaxed Bernoulli variables using $\hat{\mathbf{z}} = \boldsymbol{\sigma}(\beta l^K)$, and insert them into the probability mass function for the autoregressive discrete approximate posterior (Eq. 8). While this can be computed efficiently in parallel, it introduces another level of mismatch between the distribution we sample and the distribution under which we evaluate the probability, in the sense that we sample using the fixed point iterations, but evaluate probabilities under the sequential autoregressive process (Eq. 8). Remarkably, this is often not an issue for the discrete model, as we can simply iterate until convergence, and convergence is very well-defined as the latents are either $0$ or $1$. However, for the relaxed model it is more difficult to define convergence as the latents can lie anywhere between $0$ and $1$, so in practice we used a fixed number of iterations (in particular, $K = 5$, which was generally sufficient for the discrete model to converge).

## 2.3 Evaluating probabilities under the continuous model

To evaluate the probability of the continuous variables under the fixed point iterations, we interpret the iterations as constituting a normalizing flow. Normalizing flows exploit the fact that we can compute the probability density of a random variable, $l^k(\eta)$, generated by transforming, via a one-to-one function, a sample $\eta$,

$$\mathrm{P}\left(l^k(\eta)\right) = \mathrm{P}\left(\eta\right) \left|\frac{\partial l^k(\eta)}{\partial \eta}\right|^{-1} \tag{11}$$

where $|\partial l^k / \partial \eta|$ is the absolute value of the determinant of the Jacobian of $l^k(\eta)$. While the determinant of the Jacobian is often very difficult to compute, we can ensure the Jacobian is 1 by using a restricted family of transformations, under which the value of $l_t^k$ depends only on the current value, $\eta_t$ via simple addition, and on past values of $\eta$ via an arbitrary function,

$$l_t^{k+1} = \eta_t + g^{k+1}\left(\mathbf{x}, \eta_{1:t-1}\right). \tag{12}$$

And we know that our fixed-point iterations indeed lie in this family of functions, as Eq. (10) mirrors the above definition, with,

$$g^{k+1}\left(\mathbf{x}, \eta_{1:t-1}\right) = \mathbf{v}_t\left(\mathbf{x}, \sigma\left(\beta l_{1:t-1}^k\right)\right) \tag{13}$$

where $l_{1:t-1}^k$ indeed depends only on $\eta_{1:t-1}$ (see Eq. 10).

## 2.4 Issues with continuous model

Using continuous latent variables might seem appealing, because we can compute the exact approximate posterior probability even when the fixed point iterations have not converged, and because there is additional flexibility in that we are not restricted to approximate posteriors which have an interpretation as a discrete dynamical system. However, moving from a discrete latent variable to a continuous latent variable introduces scope for considerable mismatch between the true and approximate posterior (Fig. 1B), which simply is not possible for the binary model where the posterior remains Bernoulli. This mismatch between the approximate and true posterior implies a discrepancy between the ELBO and the true model evidence (Fig. 1C), and this discrepancy grows as the evidence in favour of $\bar{z} = 0$ or $\bar{z} = 1$ increases. This mismatch has a potentially large magnitude (compare the scale on Fig. 1B to that in Fig. 2F), and thus can dramatically modify the variational inference objective function, introducing the possibility for biased inferences. In fact, this is exactly what we see (Fig. 1D), with the approximate posterior underestimating the evidence in favour of $\bar{z} = 0$ or $\bar{z} = 1$.

Further, the Logistic approximate posterior can introduce additional biases when optimized in combination with a relaxed binary variables (Figs. 1B–D use a hard-thresholding or equivalently, an infinite inverse-temperature). In particular, we consider a Gaussian likelihood, $\mathrm{P}\left(x = 0.5|\hat{z}\right) = \mathcal{N}\left(x = 0.5; \hat{z}, \sigma\right)$, and thus, as $\sigma$ decreases, there is increasing evidence that $\hat{z}$ is close to $0.5$. Under the true model, with a hard-threshold, this can never be achieved: $\bar{z}$ must be either $0$ or $1$. However, if we combine a relaxation (Eq. 4) with a Logistic approximate posterior, it is possible to reduce the variance of the logistic sufficiently such that the relaxed Bernoulli variable, $\hat{z}$, is indeed close to $0.5$. To quantify this effect, we consider how the standard deviation of $\hat{z}$ (which should be $0.5$ under the true posterior), varies as we change the standard deviation of the likelihood, $\sigma$ (Fig. 1E).

## 2.5 Experiments

One case where binary latent variables are essential is that of calcium spike deconvolution: inferring latent binary variables representing the presence or absence of a neural spike in a small time bin, based on noisy optical observations (Fig. 2C).

We take the binary variables, $z_{it}$ as representing whether neuron $i$ spiked in time-bin $t$, and as such, we can interpret $u_{it}$ as the corresponding synaptic input (in contrast, as $\mathbf{v}_t$ is not part of the generative model, it does not have a specific a biological interpretation). For the generative model, we take a weighted sum of inputs from past spikes, filtered by a temporal kernel $\kappa^{\mathrm{u}}$ and $\kappa^{\mathrm{v}}$, which

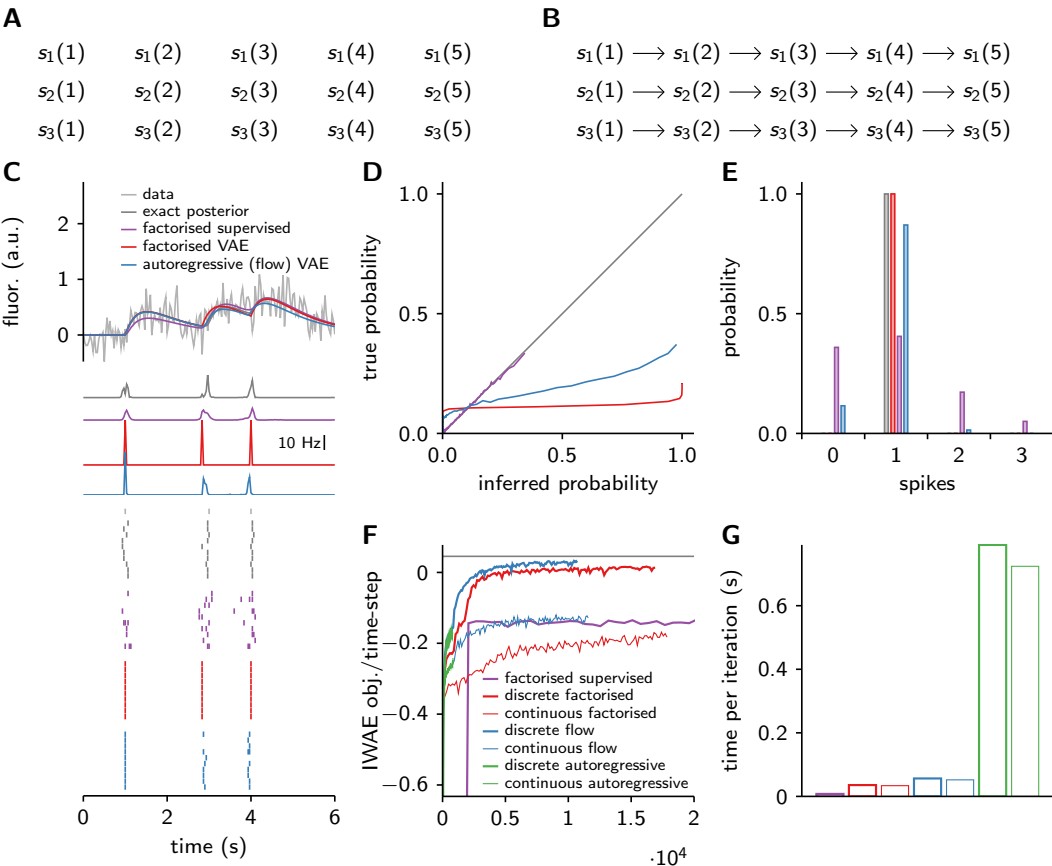

Figure 2: Correlated approximate posteriors in a single-channel (neuron) model. **A** Factorised generative model. **B** Correlated approximate posterior, with an autoregressive temporal structure, but without correlations between cells. **C** Example observed fluorescence trace and average reconstructions (top), inferred rates (middle) and inferred spiking (bottom), for the true posterior (dark gray), then models with factorised posteriors trained using VAE (red) and supervised (purple) procedures, and finally our new discrete flow trained using VAE (blue). **D** The true marginal probability of there being a spike in simulated data, against inferred probability. The optimal is unity (dark-gray). **E** The number of inferred spikes, given only one spike in the underlying data. **F** The time course of the VAE objective under the different models, and the highest possible value for the objective, estimated using importance sampling. **G** The time required for a single iteration of the algorithm, for the different variants in **F**.

is mirrored by the recognition model, to ensure that the recognition model can capture any prior-induced statistical structure,

$$\mathbf{u}_t = \mathbf{b}^{\mathrm{u}} + \mathbf{W}^{\mathrm{u}} \sum_{t'=1}^{\tau} \boldsymbol{\kappa}_{t'}^{\mathrm{u}} \odot \mathbf{z}_{t-t'} \qquad\qquad \mathbf{v}_t = \mathbf{b}^{\mathrm{v}}(\mathbf{x}) + \mathbf{W}^{\mathrm{v}} \sum_{t'=1}^{\tau} \boldsymbol{\kappa}_{t'}^{\mathrm{v}} \odot \mathbf{z}_{t-t'}. \qquad (14)$$

where $\odot$ is the Hadamard product, so $\mathbf{r} = \mathbf{g} \odot \mathbf{h}$ implies $r_i = g_i h_i$.

In the first experiment, we considered a single neuron, whose spiking was IID (Poisson) with firing rate 0.25 Hz (Fig. 2A). Fluorescence data was simulated by convolving spikes with a double-exponential temporal kernel with rise time 0.3 s and decay time 1 s and adding noise with standard deviation $e^{-1.5}$. We learned the recognition model, which consisted of two components. The first is a neural network mapping data to spike inferences, $\mathbf{b}^{\mathrm{v}}(\mathbf{x})$, which consisted of two hidden layers, with 20 units per cell per time bin, where the first layer takes input from 200 time points from a single cell's fluorescence trace, and the second layer takes 5 time points from the previous hidden layer, and we use Elu nonlinearities (Clevert et al., 2015) (for further details see Speiser et al.,

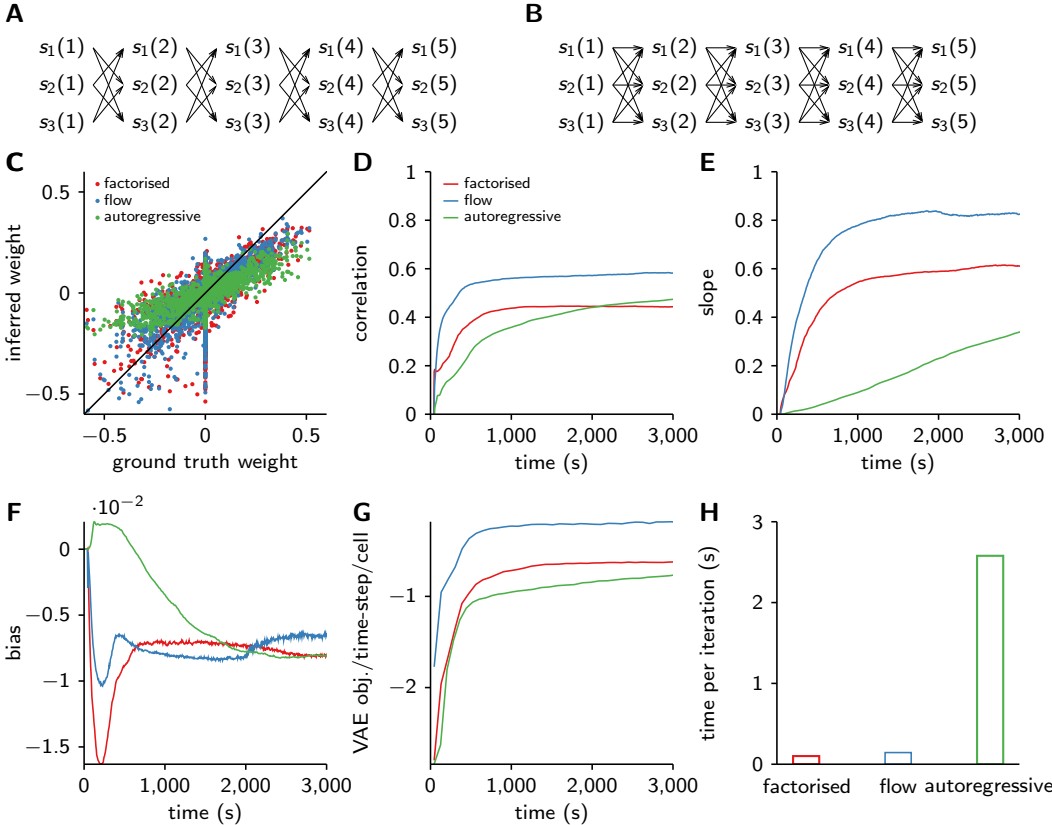

Figure 3: Correlated approximate posteriors to estimate the connectivity between multiple neurons. **A** Autoregressive generative model, incorporating the effects of synaptic connectivity (note the lack of self-connections). **B** Correlated autoregressive approximate posterior. **C** The inferred and ground truth weights using factorised, flow and autoregressive posteriors. **DEF** The correlation (**D**), slope (**E**) and bias (F) for the points in **C**, plotted across training time. **G** The time course of the VAE objective under the different models. **H** The time required for a single training iteration of the algorithm, for the different models.

2017; Aitchison et al., 2017). The second is the recognition temporal kernel $\kappa^v$, which captures the anticorrelations induced by explaining away (Fig. 2BC).

All strategies, including factorised and autoregressive (flow) VAE's, and supervised training give roughly similar reconstructions (Fig. 2C top). Thus, to understand how the autoregressive posterior is superior, we need to look in more depth at the posteriors themselves. In particular, the factorised VAE had very narrow posteriors, spuriously indicating a very high degree of certainty in the spike timing (Fig. 2C middle), whereas both the supervised and autoregressive VAE and also the true posterior (estimate by importance sampling) indicate a higher level of temporal uncertainty. These differences are even more evident if we consider spike trains sampled from the approximate posterior (Fig. 2C bottom), or if we consider calibration: the probability of there actually being a spike in the underlying data, when the inference method under consideration indicates a particular probability of spiking (Fig. 2D). However, the sampled spike trains indicate another issue: while the true and VAE posteriors generally have one spike corresponding to each ground-truth spike, supervised training produces considerable uncertainty about the spike-count (Fig. 2E). Thus, the VAE with an autoregressive (flow) posterior combines the best of both worlds: achieving reasonable timing uncertainty (unlike the factorised VAE), whilst achieving reasonable spike counts (unlike supervised training).

As such, the autoregressive VAE performs more effectively than the factorised methods considering the IWAE objective (with 10 samples) (Burda et al., 2015), under which the models are trained

(Fig. 2F). As expected given Sec. 2.4, to get good performance, it is important to compute probabilities by putting the relaxed variables into the discrete pmf (binary, as opposed to logistic), and to use the parallel fixed point iteration based flow, as opposed to the autoregressive distribution (note however, that we did not thoroughly explore the space of normalizing flows). The considerable differences in speed between the approximate parallel fixed point iterations, and the exact autoregressive computation arises from an order-of-magnitude difference in the time required for an individual iteration (Fig. 2G).

In our second experiment, we considered inferring synaptic connectivity between cells, based on noisy observations. Here, we considered a network of $100$ cells with no self-connectivity, and with weights, $\mathbf{W}^{\mathrm{u}}$ that are sparse (probability of $0.1$ of being non-zero), with the non-zero weights drawn from a Gaussian with standard deviation $5$ (Fig. 3A), and with a $200$ ms temporal decay. We used an autoregressive, fully connected recognition model (Fig. 3B). We use the same parameters as in the above simulation, except that we use a somewhat more realistic fluorescence rise-time of $100$ ms (previously, we used $300$ ms so as to highlight uncertainty in timing).

We inferred weights under three methods: a factorised VAE, and an autoregressive posterior where we use the fast-flow like sampling procedure (flow), and where we use the slow sequential sampling procedure (autoregressive) (Fig. 3C). The flow posterior gave a considerable advantage over the other methods in terms of the correlation between ground truth and inferred weights (Fig. 3D), and in terms of the slope (Fig. 3E), indicating a reduction in the bias towards underestimating the magnitude of weights, while the bias (i.e. the additive offset in Fig. 3C) remained small for all the methods. As such, the autoregressive posterior with flow-based sampling increased the ELBO considerably over the factorised model, or the autoregressive model with the slow sequential sampling, (Fig. 3G), and these differences again arise because of large differences in the time required for a single training iteration (Fig. 3H).

## 3 DISCUSSION

We have described an approach to sampling from a discrete autoregressive distribution using a parallel, flow-like procedure, derived by considering fixed-point iterations that converge to a sample from the underlying autoregressive process. We applied this procedure to speed up sampling from autoregressive approximate posteriors in the variational inference training loop. This allowed us to rapidly learn autoregressive posteriors in the context of neural data analysis, allowing us to realise the benefits of autoregressive approximate posteriors for single and multi cell data in reasonable timescales.

It is important to remember that while we can sample using $K$ fixed-point iterations, we can only evaluate the probability of a sample once it has converged. This mismatch introduces a level of approximation in addition to those that are typical when relaxing discrete distributions (Jang et al., 2016; Maddison et al., 2016), but we can deal with the additional approximation error in the same way: by evaluating the model using samples drawn from the underlying discrete, autoregressive approximate posterior.

Past work has used similar properties of the underlying generative model, to speed up message-passing based inference algorithms (Gonzalez et al., 2009; Domke, 2011). It is likely that their approach will be preferable when exact inference is possible albeit costly due to large tree-width/time-courses, whereas our approach will be preferable when exact inference is not possible due to long-range temporal dependencies.

Finally, our work suggests two directions for future work. First, while it is possible to use normalizing flows to define approximate posteriors for continuous state-space models, it may be difficult to know exactly which normalizing flow will prove most effective. In this context, our procedure of using fixed-point iterations may be a useful starting point. Second, we showed that while it may be possible to convert a discrete latent variable model to an equivalent model with continuous latents, this typically introduces considerable scope for mismatch between the prior and approximate posterior. However, the actual approximate posterior is relatively simple, a mixture of truncated Logistics, and as such, it may be possible to design approximate posteriors or even whole relaxation schemes that more closely match the true posterior, and indeed this may underlie the gains shown by (Vahdat et al., 2018).

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
