# OpenReview forum: "Discrete flow posteriors for variational inference in discrete dynamical systems"
_ICLR.cc/2019/Conference_

### Official Review · AnonReviewer3 · 2018-10-29
**Solid work, but the presentation is not self-contained and hard to follow**

**Rating:** 7
**Confidence:** 4

**Review:**

This paper has two main contributions:
 1) it extends normalizing flows to discrete settings (exploiting relaxation ideas from Jang et al and Maddison et al).
 2) it presents an approximate fixed-point update rule for autoregressive time-series that can exploit GPU parallelism.

Overall, I think the work is solid. Contribution 1 isn't very novel, but is useful and the authors did a good job there.

Contribution 2 seems more interesting, but is not as well studied. When is the fixed point update expected to work?
What assumptions does it imply? How does performance improve with the number of steps K? Does simulating for a finite steps emphasize the effect of early z's?
I'm a bit surprised that the authors did not attempt to study this part of their algorithm in isolation. They make a claims
but never look at this in detail.

That said, the authors do a good job showing the method "works", and figures 3F and 3G are particularly nice.
In 3G, is "autoregressive" supposed to converge to flow eventually?
Why don't the authors also use time as the x-axis in figure 2F (like 3F)?

My biggest complaint about the paper is the writing, which does not introduce and present ideas in a clear sequential
manner, making the paper hard to read. I realize ELBO is standard, but at least some description of the setup in equation 1
is warranted. What is x,z,\theta etc? Any paper should aim to be minimally self-contained. This continues throughout the paper, which does not really attempt to place the contribution in the larger literature, but rather just reports what the authors did and observed.

Some more examples:

Page 3: "so we need to evaluate \hat{Q}". This isn't defined. The authors should mention what \hat{Q} and \bar{Q} are.
Similarly for P. After a couple of passes through the paragraph, I could figure what the authors meant, but they
should introduce the notation they use.

In section 2, while defining their model, they do not mention the dimension of z_t until after equation 8
(and even here, it has to be inferred).

What is x in 6b? What is the generative model they are doing inference on?

Section 2.2: it's not clear to me how convergence is defined even in the discrete case. I feel this discussion
also really belongs to section 2.1

While I can understand what section 2.3 is trying to say, I could not really follow the notation.

I could not understand figure 1E and the associated sentence in section 2.4

What is the take-away of section 2.3 and 2.4? The authors seem to imply working with the discrete model is
better in their experiments. Maybe forewarn the reader here?

The experiments are a bit hard to follow. It is inspired by a neuroscience application, but uses only simulated data. This is fine, but rather than describe the setup in mathematical/time-series language, it is complicated the with neuroscience jargon. As such, it feels disjointed and disconnected from the rest of the paper. I already complained that earlier sections do not describe the modeling setup, this is one way the paper could be improved.

In figure 2A and 3A, are the s's actually z's?

---

### Official Review · AnonReviewer1 · 2018-11-02
**Promising idea but lack of thorough experiments**

**Rating:** 4
**Confidence:** 4

**Review:**

Normalizing flows provide a way for variational posteriors to go beyond the mean-field assumption and introduce correlations in the posterior for latent variables. Typically, normalizing flows are only defined for continuous distributions, and the authors tackle the issue of creating flexible variational posteriors for discrete latent models. They posit a general autoregressive posterior family for discrete variables or their continuous relaxations. In order to perform variational inference with reparameterization gradients, one needs to have a sample from their variational family and be able to evaluate the density at the sample. The obvious way to sample from this autoregressive variational family is O(T) for T the number of autoregressive time-steps. Instead, they propose a method based off fixed point iterations to compute logits in parallel based off the results of previous iterations to generate an approximate sample. Moreover, they can interpret each iteration as a volume-preserving flow, so they don't have to add any terms to the density. They also use a continuous relaxation of Bernoulli random variables so they can back-propagate through a recognition network. They use their method on synthetic calcium spike data, and show that the correlated posterior is better-suited to handle uncertainty.

This is certainly original work, and it presents it in a general way. The authors' formulation of the probabilistic model and variational family in equations 6a-7b can be extended to any autoregressive family. As the authors mention, they also do not need to work only with continuous relaxations of Bernoulli random variables, as they can extend it to categorical random variables using something like the Gumbel-Softmax/Concrete distributions.   Although the idea behind parallelizing updates is not inherently elaborate, the authors provide an interesting interpretation as a normalizing flow.

My main criticism of the paper is the experiments section. The experiments are only performed on synthetic data sets. How do the methods scale to larger data sets? The authors state that iterations of the fixed point procedure converge rapidly in practice, but it seems like it's only been evaluated on these synthetic data sets. It seems like performing K fixed point iterations fixes the dependencies to be in a window of size K, so this may perform worse in practice for models that have long temporal dependencies (or for non-temporal latent discrete random variables where the ordering is not important).

The experiments also do not seem to evaluate any held-out metrics. The experiments would be stronger by e.g. approximating the marginal held-out loss (perhaps using IWAE or otherwise), since it seems almost guaranteed that more flexible variational families should achieve a tighter bound on the training set (it's possible that there were actually held-out metrics but I missed them, in which case please let me know).

Another criticism of the paper is with the clarity. The authors sometime use notation before/without defining it. For example, T is used without definition it at the beginning of page 2 and N is used without definition at the beginning of section 2.1. It makes it difficult to have intuition for the math as a result of not knowing the definitions. Even things like explicitly stating that k is a timestep index before equation 9 would be helpful.

More minor notes:

-I got a lot out of Figure 1A. For Figure 1B-E, what is beta? Is there intuition for how these results change as a function of beta?

-When you say Equation 8 in section 2.2 I believe you mean equation 7b.

Overall, there has not been much (if any) work for correlated posterior families for discrete latent variable models, and the authors have provided a promising first step. The next step would be seeing more experimental results for a larger variety of models.

PROS
-Idea is very interesting and novel, with a nice connection to normalizing flows
-Underexplored area of research, promising first steps.

CONS
-Experiments should be more thorough
-Lack of clarity made it hard to understand at certain points

---

### Official Review · AnonReviewer2 · 2018-11-04
**Poorly developed (potentially useful) idea**

**Rating:** 4
**Confidence:** 3

**Review:**

This paper uses an autoregressive filtering variational approximation for parameter estimation in discrete dynamical systems. One issue that crops up with this *particular choice* of variational distribution is that (a) inference proceeds sequentially (by definition) and (b) this does not make use of parallelism in modern hardware. To mitigate this, the paper proposes using fixed point iterations. After the first iteration, the approximate posterior for each latent variable corresponds to a random draw from a logistic distribution. Each subsequent application of the fixed point iteration modifies the posterior distribution by incorporating information from the previous latent state. After T applications of the fixed point equation, the procedure approximates an auto-regressive variational posterior distribution. Its possible I've misunderstood the point being made in Sections 2.2 - 2.4, but the paper points out that the choice of iterations "looks like" a normalizing flow (with Jacobian 1).

The method for inference is evaluated on two synthetic datasets. The paper finds that the flow-based approach takes less time than using the full autoregressive variational posterior and learns less bias weights than a fully factorized approach.

Overall:
I think the idea of approximating posterior distributions via fixed point iterations as presented here is interesting since it presents a reasonable way to trade off between expressivity and computational complexity. However, in this manuscript the idea is insufficiently explored and not presented clearly.

Clarity -- methodology:
The paper is poorly written. It is formatted in an awkward manner making it quite difficult to understand what model was considered here. For example, the *first equation* in the paper is a variational lower-bound. The equation is present in the absence of describing what generative model is considered in this work (or even stating what P and Q are, and how they factorize). Unless I'm missing something, as long as the *final step* of every fixed-point iteration (at each point in time) realizes a valid prediction of the mean parameter of continuous distribution relaxed to a discrete one, the proposed method is still valid for approximate inference. Why then is the relationship to normalizing flows important to highlight or emphasize?

Clarity -- experimental results:
The baselines in the experimental section are not described. Out of the blue, one of the method describes "supervised training" (up until that point, I was under the impression that the model was entirely unsupervised). Where does the supervision comes from? The IWAE objective is mentioned without justification in the experimental section whereas the methodology section describes learning with the lower-bound.

Larger time series problems:
A reason, motivated by the paper, for considering this method was the potential to parallelize computation of the approximate posterior distribution on GPUs. Yet, the evaluation was conducted on significantly smaller problems. The paper would be strengthened by an evaluation on density estimation on larger, higher dimensional datasets [e.g. the benchmark polyphonic music dataset -- http://www-etud.iro.umontreal.ca/~boulanni/icml2012].

*Why* does convergence happen quickly?
An unanswered issue is that a central claim of the paper hinges on an empirical observation -- namely that the fixed point iterations converge "quickly". In the absence of theory on *why* and how quickly we might expect convergence and what convergence depends on, I think there is a need for further experimentation to understand and characterize situations when we can expect rapid convergence. Intuitively, the number of fixed point iterations controls how far in the past the posterior distribution for the latent variable at timestep t depends. Rapid convergence, as observed here, could happen because the experiments only consider simplistic generative models in which the true posterior distribution is well approximated using a small temporal context from the past.

---

### Meta-Review · Area_Chair1 · 2018-12-19
**Interesting original idea but unclearly presented and with a too limited experimental validation.**

**Confidence:** 4
**Recommendation:** Reject

**Metareview:**

The paper presents an original approach to replace inefficient discrete autoregressive posterior sampling by a parallel sampling procedure based on fixed-point iterations reminiscent of normalizing flow, but for discrete variables.
All reviewers liked the idea, and found that it was an original and promising approach. But all agreed the paper was poorly written and very unclear.
All also found the experimental section lacking, in clarity and scope.

Authors did not provide a rebuttal.

Overall a potentially really promising idea, but the paper is not yet ripe.